# SMAD3 rs17228212 Polymorphism Is Associated with Advanced Carotid Atherosclerosis in a Slovenian Population

**DOI:** 10.3390/biomedicines12051103

**Published:** 2024-05-16

**Authors:** David Petrovič, Jernej Letonja, Danijel Petrovič

**Affiliations:** 1Laboratory for Histology and Genetics of Atherosclerosis and Microvascular Diseases, Faculty of Medicine, University of Ljubljana, Korytkova 2, 1000 Ljubljana, Slovenia; ihe@mf.uni-lj.si (D.P.); jernej.letonja@mf.uni-lj.si (J.L.); 2Institute of Histology and Embryology, Faculty of Medicine, University of Ljubljana, Korytkova 2, 1000 Ljubljana, Slovenia

**Keywords:** carotid atherosclerosis, SMAD3, rs17228212, polymorphism

## Abstract

Smad proteins influence the TGFβ signaling pathway, which plays an important role in the progression of atherosclerosis. The aim of our study was to investigate the association between the rs17228212 polymorphism of the SMAD3 gene and advanced carotid atherosclerosis in Slovenian subjects and to investigate the effect of the rs17228212 *SMAD3* polymorphism on the expression of SMAD3 in endarterectomy sequesters. In this cross-sectional case-control study, 881 unrelated Caucasians were divided into two groups. The first group included 308 patients with advanced carotid atherosclerosis of the common or internal carotid artery with stenosis greater than 75% that underwent a revascularization procedure (cases). The control group consisted of 573 subjects without hemodynamically significant carotid atherosclerosis. We analyzed the rs17228212 polymorphism of the SMAD3 gene using the StepOne real-time polymerase chain reaction system and TaqMan SNP genotyping assay. The results in the two genetic models showed a statistically significant association, codominant (OR 4.05; CI 1.10–17.75; *p* = 0.037) and dominant (OR 3.60; CI 1.15–15.45; *p* = 0.045). An immunohistochemical analysis of SMAD3 expression was conducted for 26 endarterectomy specimens. The T allele of the rs17228212 SMAD3 gene was shown to be associated with an increased numerical area density of SMAD3-positive cells in carotid plaques.

## 1. Introduction

Atherosclerosis is a chronic inflammatory disease. Atherosclerosis is characterized by the progressive deposition of lipids, connective tissue, smooth muscle cells, cellar debris, and calcium, accompanied by inflammation of the arterial walls [1,2]. Predilection sites for atherosclerotic plaques are arterial branch points where there is turbulent flow and lipids enter and remain in the subendothelial space [1,2]. Atherosclerosis is caused by a combination of genetic and environmental risk factors. The most important risk factors are arterial hypertension, obesity, and elevated levels of low-density cholesterol [1,2]. Atherosclerosis is a major cause of mortality worldwide due to diseases such as coronary and cerebral ischemic disease [1,2].

Carotid atherosclerosis is a common cause of strokes and transient ischemic attacks (TIAs). If the inner diameter of the internal carotid arteries (ICAs) is narrowed, this can lead to a reduction in blood and oxygen supply to the brain and cause various neurological problems. Stenosis of the internal carotid artery diameter exceeding 50% is associated with 15% of cases of ischemic stroke [3].

Smooth muscle cells (SMCs) play an important role in plaque formation. Studies have shown that the majority of cells within a plaque, including those that exhibit some inflammatory markers, are derived from oligoclonal SMCs that have dedifferentiated [4,5,6,7]. Genes that affect SMCs also alter the composition of the atherosclerotic plaque [6].

SMAD3 is an intracellular messenger that regulates TGF signaling, and TGFβ signaling plays a critical role in the development of smooth muscle cells in plaques [8,9]. The gene for SMAD3 is located at the chromosomal locus 15q22.33. The polymorphism rs17228212 is an intronic SNP, the T allele is the ancestral allele and the frequency of the C allele (MAF) is 29% in the European population. TGFβ is also involved in the inflammatory response, epithelial–mesenchymal transition and embryonic development [10]. Under most conditions, it inhibits proliferation and migration in the extracellular space [11,12,13]. It also inhibits the formation of foam cells [14]. These in vitro activities support the hypothesis that TGFβ plays a role in maintaining the normal architecture of blood vessels. The administration of anti-TGF-β1 antibodies results in increased vascular inflammation and accelerated lipid lesion formation, and increases plaque instability in animals [15]. Additionally, in rodents, strategies that increase TGFβ activity protect against the atherogenic changes in the vessel wall [16]. On the other hand, a number of studies have shown that the extent of atherosclerosis correlates with the degree of endothelial-to-mesenchymal transition activated by TGFβ signaling [17,18,19].

Smad proteins influence the TGFβ signaling pathway, particularly SMAD2 and SMAD3. Although they do not bind effectively to DNA on their own, they interact with other transcription factors and play a central role in regulating essential transcriptional programs that control cell developmental processes and disease [20,21,22].

The SMC-specific deletion of *SMAD3* has been shown to affect dedifferentiated SMC in atherosclerotic plaques [23]. It alters SMC towards a pro-remodeling phenotype characterized by the activation of genes associated with tissue remodeling, such as Mmp3, and the production of inflammatory chemokines such as Cxcl12 [23]. It also leads to an increase in SMC-derived chondromyocytes (CMC) [23]. These changes lead to increased outward remodeling and calcification within the plaques, which are normally inhibited by SMAD3 and its effect on Hox and Sox factors [23].

Although the mechanism of action of SMAD3 is quite well understood, there are not many genetic association studies in the field of atherosclerosis [24,25,26]. We selected the rs17228212 polymorphism because the minor allele frequency was >10% in the European population.

The aim of our study was to investigate the association between the rs17228212 polymorphism of the SMAD3 gene and advanced carotid atherosclerosis in Slovenian subjects. The second aim was to investigate the effect of the rs17228212 *SMAD3* polymorphism on the expression of the SMAD3 in the endarterectomy specimens obtained during carotid endarterectomy.

## 2. Materials and Methods

### 2.1. Patients

In this cross-sectional case-control study, 881 unrelated Caucasians were divided into 2 groups. This cross-sectional case-control study was conducted according to the STROBE guidelines. The first group included 308 patients with advanced carotid atherosclerosis and a hemodynamically significant narrowing of the common carotid artery or internal carotid artery (>75% narrowing of the lumen) who underwent a revascularization procedure (endarterectomy or stent implantation). The control group consisted of 573 subjects who did not have advanced atherosclerosis (<50% narrowing of the lumen) in their CCA or ICA or a history of stroke/transient ischemic attack (controls).

The control group included people who came for a routinely scheduled cardiovascular risk assessment. We included individuals of both sexes without symptomatic carotid artery disease. These individuals had no ultrasound-detectable atherosclerotic changes or mild atherosclerotic changes in their carotid arteries. The extent of the stenosis in the common carotid artery (CCA) or the internal carotid artery (ICA) had to be hemodynamically insignificant, i.e., less than 50%. The group of cases included patients who had a hemodynamically significant stenosis of the CCA or ICA (>75%) and underwent endarterectomy or stent implantation.

Cases and control subjects were recruited from 3 healthcare facilities in Slovenia: Cardiovascular Center Medicor d.d. Ljubljana, University clinical centre Maribor and General Hospital Izola. The recruitment process took place over a period of 10 years, from 2010 to 2019.

Specialized medical professionals from the aforementioned healthcare facilities performed ultrasound examinations and, if necessary, CT angiographies. Ultrasound was used to measure the thickness of the intima media in the common carotid artery (CCA), the internal carotid artery (ICA), the external carotid artery (ECA) and the vertebral artery (VA) on both sides. The examinations also included measuring the internal diameter of the ICA, the blood flow rate, and the presence, type and size of atherosclerotic plaques. The thickness of the intima media on both sides of the ICA was determined by taking the average of three measurements.

General information and a history of risk behavior, as well as clinical and laboratory parameters such as age, gender, systolic and diastolic blood pressure, alcohol consumption, smoking habits, fasting glucose levels, body mass index ((BMI), total cholesterol, high-density lipoprotein cholesterol (HDL-C), low-density lipoprotein cholesterol (LDL-C), triglycerides, high-sensitivity C-reactive protein (CRP), coronary artery disease (CAD), myocardial infarction (MI), glycated hemoglobin (HbA1c), statin treatment information, duration of arterial hypertension (AH) and diagnosis of type 2 diabetes mellitus (T2DM), were carefully recorded in the medical records.

The exclusion criteria were as follows: carotid stenosis of non-atherosclerotic origin, patients with aortic arch stenosis or stenosis of the right subclavian artery, patients with neck tumor and patients with incomplete data. The study was designed in accordance with the Declaration of Helsinki and approved by the Slovenian National Medical Ethics Committee (number 0120-316/2023). A detailed interview and physical examination were performed only after participants signed an informed consent.

### 2.2. Genotyping

Deoxyribonucleic acid (DNA) was isolated from peripheral blood leukocytes in the laboratory of the Institute of Histology and Embryology, Faculty of Medicine, University of Ljubljana. A total of 3–12 μg of genomic DNA was isolated from 200 μL of blood using a QIAcube device (Qiagen GmbH, Hilden, Germany) according to the “V3” protocol. In this procedure, a commercially available QIAamp DNA Blood Mini Kit (250) (Qiagen GmbH, Hilden, Germany) was used, consisting of five different reagents (AL buffer, 96% ethanol, AW1 buffer, AW2 buffer, AE buffer) and an appropriate amount of protease (Qiagen GmbH, Hilden, Germany) at a ratio of 285 μL per 200 μL of blood. The rs17228212 polymorphism was genotyped in both groups (cases and controls) according to the manufacturer’s recommendations using the TaqMan SNP Genotyping Assay (Applied Biosystems, Foster City, CA, USA).

### 2.3. Immunohistochemistry

Immunohistochemical analysis was performed on 26 endarterectomy specimens obtained from 26 participants with advanced carotid atherosclerosis during carotid surgery. Formalin-embedded (FFPE) tissue sections of the internal carotid artery (ICA) were subjected to hematoxylin–eozin staining. Consecutive 5 μm tissue sections were cut, mounted and dried on glass slides from each paraffin block. Paraffin blocks were deparaffinized and dehydrated in graded alcohol solutions.

The Novo Link Max Polymer Detection System (Leica Biosystems Newcastle Ltd., Newcastle upon Tyne, United Kingdom) was utilized for the detection of SMAD3-positive cells. Anti-SMAD3 monoclonal antibodies (Thermo Fisher Scientific, Waltham, MA, USA), at a dilution of 1:100, were applied overnight at 4 °C. Placental tissue was used as a positive control and tonsil tissue as a negative control. We defined the cells as SMAD3-positive/negative. The area containing SMAD3-positive cells was manually marked, and the numerical area density of SMAD3-positive cells was calculated as the number of positive cells per square millimeter [27].

### 2.4. Statistical Analysis

The statistical analysis was carried out using SPSS software for MS Windows, version 26.0 (IBM, North Castle, NY, USA). The Shapiro–Wilk test was used to test for normal distribution. Normally distributed continuous clinical data are presented as mean ± SD, while non-normally distributed data (continuous variables) are presented as median and interquartile range. For categorical variables, the number and percentage of affected patients were reported. Continuous clinical parameters that were normally distributed were compared using the unpaired Student’s *t*-test, and the Mann–Whitney U test was used to examine non-normally distributed continuous data. Discrete variables were compared using the chi-square test. A logistic regression analysis was performed to examine variables that showed significant differences in the univariate analysis (*p* < 0.05). The deviation from the Hardy–Weinberg equilibrium (HWE) was analyzed using the chi-square goodness of fit test.

## 3. Results

Table 1 shows the clinical characteristics and biochemical parameters of the cases with advanced carotid atherosclerosis (subjects with ICA stenosis > 75) and controls (subjects without hemodynamically significant carotid atherosclerosis). There were no statistically significant differences in systolic blood pressure, HDL cholesterol, triglycerides and HbA1c. The cases were older, and had a lower BMI, higher waist circumference, lower diastolic blood pressure, lower fasting blood glucose, lower total cholesterol, lower LDL cholesterol and higher CRP. In addition, the incidence of smoking was lower in cases and there was a lower incidence of T2DM in cases in comparison with controls. More participants in the case group were prescribed statins than those in the control group.

The distribution of the genotype and allele frequency of the rs17228212 polymorphism of the SMAD3 gene in the case and control groups is shown in Table 2. To determine the genotype distributions of cases and controls, the data were brought into Hardy–Weinberg equilibrium (cases: *p* = 0.9205, controls: *p* = 0.2910, Pearson χ^2^ test, respectively). Through univariate analysis, we found significant differences in genotype (*p* = 0.033) and allele distribution (*p* = 0.009) between cases and controls. The TT genotype was more common in cases, whereas the CC genotype was more common in controls. In addition, the T allele was more common in cases and the C allele was more common in controls.

We used logistic regression to determine whether the rs17228212 polymorphism was associated with advanced carotid atherosclerosis after adjusting for age, BMI, waist circumference, diastolic blood pressure, fasting glucose, cholesterol, sex, smoking, T2DM and statin treatment. The results in the two genetic models showed a statistically significant association, both codominant (OR 4.05; CI 1.10–17.75; *p* = 0.037) and dominant (OR 3.60; CI 1.15–15.45; *p* = 0.045). In the recessive genetic model, there was no statistically significant association (OR 1.63; CI 0.91–2.95; *p* = 0.10) (Table 3).

In the endarterectomy sequesters from surgically treated patients with advanced carotid atherosclerosis, there was an increase in the numerical areal density of SMAD3-positive cells in subjects with the T allele in comparison to those with the C allele (41 ± 6/mm^2^ versus 25 ± 4/mm^2^; *p* < 0.001) (Figure 1). Photographs of all 26 endarterectomy sequesters that were immunohistochemically stained for SMAD3 are presented in Figure 2.

## 4. Discussion

Our cross-sectional case-control study demonstrated an association between the T allele of rs17228212 polymorphism and advanced carotid atherosclerosis. Homozygotes for the T allele (wild type) were 4.05 times more likely to have advanced carotid atherosclerosis than the homozygotes for the C allele according to the co-dominant genetic model (*p* = 0.037). According to the dominant genetic model, both TT homozygotes and the heterozygotes were 3.60 times more likely to have advanced carotid atherosclerosis (*p* = 0.045).

The T allele was also associated with an increase in the numerical areal density of SMAD3-positive cells in endarterectomy sequesters. We found no other published articles investigating the influence of the rs17228212 polymorphism on the expression of SMAD3 in atherosclerosis. Miller et al. reported an association between another *SMAD3* polymorphism (rs17293632) and increased CAD risk. They also reported an association between the risk allele C of the rs17293632 polymorphism and an increase in the gene expression of SMAD3 in human arterial smooth muscle cells [28].

Participants in the control group had a higher prevalence of T2DM and had higher levels of total cholesterol, LDL-cholesterol (a more proatherogenic lipid status) and glucose in comparison with the participants in the case group. The average participant in both groups had hypertension; however, there was no statistically significant difference between the groups. A possible explanation for this is that the participants in the case group were more adherent to their therapy. The majority of participants in both the case group (85.4%) and control group (74.7%) were prescribed statins (*p* < 0.001). The difference between the groups was statistically significant and could partially explain the more atherogenic lipid status in the control group. A total of 14.6% of the participants in the case group were not prescribed statins. These patients either had statins replaced with other medication or were prescribed statins after the visit to the healthcare facility in which they were recruited for this study.

Carotid atherosclerosis can lead to stroke or transient ischemic attack. It is caused by the progressive accumulation of plaques in the arterial walls. The majority of the cells that make up a plaque originate from dedifferentiated SMCs [4,5,6,7].

There are three functional subfamilies of SMAD3. Receptor-activated Smads (R-SMADS: SMAD1, SMAD2, SMAD3, SMAD5 and SMAD8), which are phosphorylated by the type I receptor, inhibitory SMADS (I-SMADS: SMAD6 and SMAD7) that provide negative feedback by labeling receptors for decay and competing with R-Smads for interaction with receptors, and common mediating SMADS (Co-SMADS: SMAD4) that oligomerize with activated R-SMADS [29].

The deletion of *SMAD3* in SMC alters de-differentiated SMC in atherosclerotic plaques [23]. This causes the SMC to switch to a pro-remodeling phenotype and produce increased amounts of inflammatory chemokines (Cxcl12). In addition, studies have shown that genes associated with tissue remodeling, such as Mmp3, are activated [23]. There is also an increase in the SMC-derived chondromyocyte (CMC) population [23]. The aforementioned changes are normally inhibited by SMAD3 and its effect on Hox and Sox; otherwise, they promote increased outward remodeling and calcification of the plaques [23]. Overall, this evidence is consistent with the theory that SMAD3 is an intracellular messenger that promotes TGF signaling [30,31,32].

Mercedes Garcia-Bermúdez et al. investigated the influence of the *SMAD3* rs17228212 polymorphism on cerebral vascular accidents and subclinical carotid atherosclerosis in RA patients negative for anti-CCP antibodies [24]. They found that the presence of the C allele of the *SMAD3* rs17228212 polymorphism was associated with a protective effect against the risk of cerebrovascular outcomes [24]. Our results are in accordance with theirs: the C allele reduces the risk of developing atherosclerosis and the T allele increases it.

Samani et al. reported an association between the *SMAD3* rs17228212 polymorphism and coronary heart disease in a GWAS study [25]. In this large study (Wellcome Trust Case Control Consortium, involving 1926 patients with CAD and 2938 control subjects, replicated in the German Myocardial Infarction Family Study, involving 875 patients with myocardial infarction and 1644 control subjects), Samani and coworkers reported that the C allele of the SMAD3 gene (located at the locus 15q22.33 (rs17228212)) influences the risk of developing coronary artery disease [26]. The SMAD3 gene locus 15q22.33 (rs17228212) and genes in some other loci (*PSRC1* at 1p13.3, *MIA3* at 1q41) play a role in cell growth or inhibition [26,27,29]. The process of cell growth inhibition is thought to be fundamental to the formation and progression of atherosclerotic plaque and also to plaque instability [30].

On the other hand, in a much smaller Iranian study (101 CHD patients with a luminal stenosis of ≥ 50% of any coronary vessel and 111 healthy individuals), Rayat et al. found no association between the rs17228212 and the risk of coronary artery disease [26].

The results of the studies performed by Samani et al. and Rayat et al. differ from our conclusions. The study performed by Rayat et al. was conducted on an Iranian population and a smaller sample, which could explain the different conclusions. Both studies also investigated the association between the rs17228212 polymorphism and coronary artery disease, which might explain the differing conclusions.

The TGFβ signaling pathway plays an important role in regulating the inflammatory response as well as the progression of smooth muscle cells in plaques [8,9]. SMAD3 acts as a regulating factor of the intracellular messenger of the TGF signaling pathway [8,9]. The ligands that contribute to the progression of atherosclerosis in the TGFβ superfamily are mainly TGFβ (TGF-β1, TGF-β2 and TGF-β3) and bone morphogenetic protein (BMP) [10,33,34]. After the ligand binds to the receptor (TGF-βRII), they form a heteromeric complex, the SMAD anchor for receptor activation, which recruits specific receptor-regulated Smad (R-Smad) that binds to TGF-βRI and is phosphorylated [10,33,34]. R-Smad separates from the original complex and joins with a common partner Smad (Co-Smad) to form a new complex [10,33,34]. This new complex interacts and migrates into the nucleus and regulates the activation of specific genes by affecting their transcription [10,33,34]. SMAD3 regulates macrophage activation in renal inflammation and also regulates the expression of miR-21, which is involved in inflammation associated with atherosclerosis [35,36]. However, our cross-sectional study has some limitations. It was conducted in an ethnically homogeneous cohort from Slovenia, and the cohort was relatively small. The second limitation in our study was the differences in various parameters (age, gender, BMI, etc.). We minimized their influence by including all these parameters in the logistic regression models. Moreover, we included only one polymorphism of the SMAD3 gene, so we cannot exclude the influence of other polymorphisms. Also, our endarterectomy sequester sample was relatively small, which could have impacted our results. Therefore, research on larger and more ethnically diverse subjects is needed to improve our understanding of the role of SMAD3 in the development of atherosclerosis.

We have to acknowledge that, currently, only the multifactorial/multiSNPs approach (with a reliable polygenic risk score) can be generally recommended [37,38]. However, the results of our study suggest that the rs17228212 polymorphism may be used in the assessment of risk for advanced carotid atherosclerosis in a clinical setting, not only in patients with established advanced atherosclerotic disease but also in individuals who do not have manifest atherosclerosis. In addition, we might consider its potential applicability in close relatives of patients with advanced carotid disease. Confirmation of the presence of the rs17228212 polymorphism may guide the adaptation of treatments to a more proactive approach. This includes more rigorous counseling and monitoring, particularly regarding therapeutic lifestyle changes. This could also include stricter dietary adjustments to reduce systemic proinflammatory conditions. In addition, the earlier achievement of recommended blood pressure levels and striving for the lowest possible levels of atherogenic blood lipids, especially LDL-C, could be emphasized. However, we need further studies with larger samples to demonstrate the necessary validity of the predictive value of the rs17228212 polymorphism as a true modifier of the risk for advanced atherosclerotic cardiovascular disease.

## 5. Conclusions

In our cross-sectional case-control study, we showed that the *SMAD3* rs17228212 polymorphism was associated with advanced carotid artery disease in a Slovenian cohort. Since SMAD3 is a part of the TGFβ signaling pathway and most studies have shown that activation of the TGFβ signaling pathway is anti-atherogenic, we hypothesized that the T allele of *SMAD3* rs17228212 may play a role in the development of atherosclerosis, suppress the expression of SMAD3, inhibit the TGFβ pathway and therefore lead to pro-atherogenic changes. Our hypotheses should be further tested with more extensive and larger studies on different polymorphisms and drugs affecting SMAD3 gene expression.

## Figures and Tables

**Figure 1 biomedicines-12-01103-f001:**
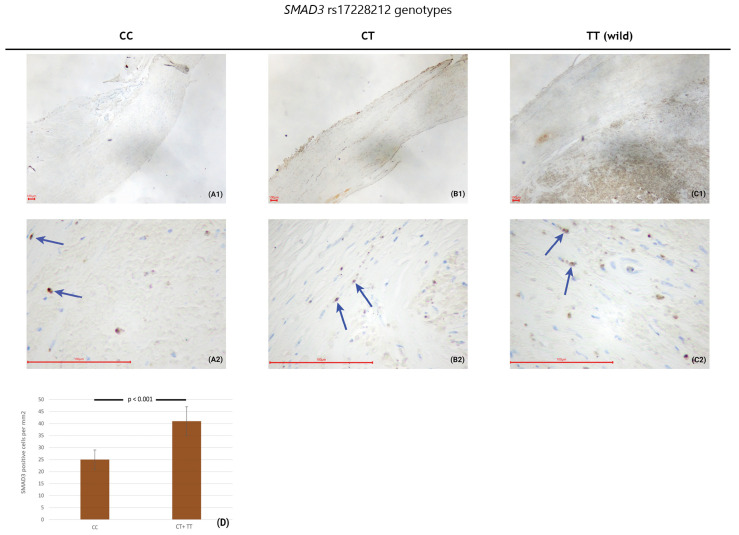
Immunohistochemical staining of endarterectomy sequester samples relative to the *SMAD3* rs17228212 genotypes. The pictures were taken at two different magnifications: the top row was taken at 4× and the bottom row at 40× magnification. SMAD3-positive cells are stained brown, SMAD3 negative cells are blue. We marked some SMAD3-positive cells with blue arrows on the pictures taken at 40× magnification. (**A1**) Endarterectomy sequester, immunohistochemically stained for the SMAD3 of a participant with the CC genotype at 4× magnification; (**A2**) Endarterectomy sequester, immunohistochemically stained for the SMAD3 of a participant with the CC genotype at 40× magnification; (**B1**) endarterectomy sequester, immunohistochemically stained for the SMAD3 of a participant with the CT genotype at 4× magnification; (**B2**) endarterectomy sequester, immunohistochemically stained for the SMAD3 of a participant with the CT genotype at 40× magnification; (**C1**) endarterectomy sequester, immunohistochemically stained for the SMAD3 of a participant with the TT genotype at 4× magnification; (**C2**) endarterectomy sequester, immunohistochemically stained for the SMAD3 of a participant with the TT genotype at 40× magnification; (**D**) bar graph comparing the numerical areal density of SMAD3-positive cells in patients with the CC genotype with those with the CT + TT genotypes (the scale bar represents 100 µm).

**Figure 2 biomedicines-12-01103-f002:**
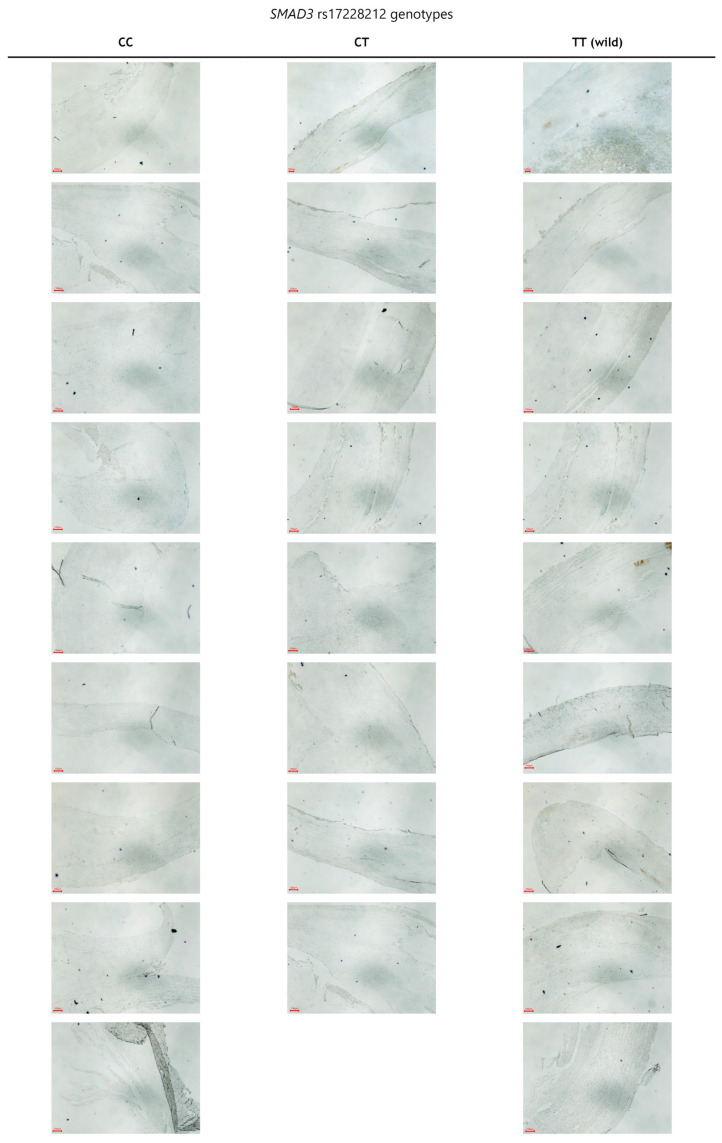
Immunohistochemical staining of all 26 endarterectomy sequester samples relative to the *SMAD3* rs17228212 genotypes. All the images are at 4× magnification. (The scale bar represents 100 µm).

**Table 1 biomedicines-12-01103-t001:** Clinical and laboratory characteristics of cases and controls.

	Case (N = 308)	Control (N = 573)	*p*-Value
Age (years)	70.94 ± 8.39	65.48 ± 11.20	<0.001
Bmi (kg/m^2^)	28.02 ± 4.18	28.90 ± 4.21	0.010
Waist circumference (cm)	101.58 ± 12.25	97.28 ± 13.87	<0.001
SBP (mm Hg)	145.88 ± 20.49	147.93 ± 21.37	0.23
DBP (mm Hg)	80.36 ± 10.55	83.75 ± 10.61	<0.001
Fasting glucose (mmol/L)	6.77 ± 2.37	7.35 ± 2.75	0.008
Total cholesterol (mmol/L)	4.49 (3.70–5.30)	4.70 (4.00–5.60)	<0.001
HDL-cholesterol (mmol/L)	1.30 (1.10–1.50)	1.31 (1.00–1.50)	0.71
LDL-cholesterol (mmol/L)	2.40 (1.90–3.20)	2.70 (2.10–3.40)	0.008
TGS-cholesterol (mmol/L)	1.40 (1.00–1.90)	1.40 (1.00–2.10)	0.44
HbA1c (%)	7.50 (7.05–9.03)	7.40 ± (6.62–8.20)	0.078
hs CRP (mg/L)	3.30 (2.90–7.00)	2.00 (1.00–3.98)	<0.001
Gender			<0.001
Male	209 (67.9%)	302 (52.7%)	
Female	99 (32.1%)	271 (47.3%)	
Smoking (%)			<0.001
Never + former smokers	228 (74.0%)	507 (88.5%)	
Active smoker	80 (26.0%)	66 (11.5%)	
DM2			<0.001
Yes	131 (42.5%)	348 (60.7%)	
No	177 (57.5%)	225 (39.3%)	
Statin treatment			<0.001
Yes	263 (85.4%)	428 (74.7%)	
No	45 (14.6%)	145 (25.3%)	

**Table 2 biomedicines-12-01103-t002:** Distribution of rs17228212 polymorphism genotypes and alleles.

SMAD3 rs17228212	Case (N = 308)	Control (N = 573)	*p* Value
TT	197 (64.0%)	323 (56.4%)	0.033
CT	99 (32.1%)	208 (36.3%)	
CC	12 (3.9%)	42 (7.3%)	
ALLELES			
T	493 (80.0%)	854 (75.5%)	0.009
C (MAF)	123 (20.0%)	292 (25.5%)	
HWE (pvalue)	0.9205	0.2910	
DOMINANT			
TT + CT	296 (96.1%)	531 (92.7%)	0.043
CC	12 (3.9%)	42 (7.3%)	
RECESSIVE			
TT	197 (64.0%)	323 (56.4%)	0.029
CT + CC	111 (36.0%)	250 (43.6%)	

**Table 3 biomedicines-12-01103-t003:** Logistic regression analysis adjusted for different variables according to genetic models.

SMAD3 rs17228212	Count	OR (95% CI)	*p* Value for OR
co-dominant			
TT vs. CC	197/323 vs. 12/42	4.05 (1.10–17.75)	0.037
CT vs. CC	99/208 vs. 12/42	2.88 (0.75–13.07)	0.14
dominant			
[TT + CT] vs. CC	296/531 vs. 12/42	3.60 (1.15–15.45)	0.045
recessive			
TT vs. [CT + CC]	197/323 vs. 111/250	1.63 (0.91–2.95)	0.10

Adjusted for age, BMI, waist circumference, diastolic blood pressure, fasting glucose, cholesterol, gender, smoking, DM2 and statin treatment.

## Data Availability

The data presented in this study are available on request from the corresponding author due to sensitive information (patients’ clinical data).

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
