# Peer review of "SMAD3 rs17228212 Polymorphism Is Associated with Advanced Carotid Atherosclerosis in a Slovenian Population"

_biomedicines, 2024, doi:10.3390/biomedicines12051103_

Round 1

Reviewer 1 Report

Comments and Suggestions for Authors

The authors have performed analysis to establish the relationship between SNP rs17228212 with SMAD3 expression and carotid atherosclerosis stage. The following comments could be addressed to further improve the manuscript.

1.      Qpcr of Smad3 could be performed to in carotid atherosclerotic plaque tissue if samples are available to help validate the genotypes and smad3 expression.

2.      Which part of the plaque are we looking at in Figure 1? It is helpful to infer the function of SMAD3 in certain cell types.

3.      Statistical analysis of Figure 1 results should be presented as well. It is important to mention how many patient samples are included for SMAD3 staining and what stages of carotid atherosclerosis they are at.

Author Response

The authors have performed analysis to establish the relationship between SNP rs17228212 with SMAD3 expression and carotid atherosclerosis stage. The following comments could be addressed to further improve the manuscript.

Qpcr of Smad3 could be performed to in carotid atherosclerotic plaque tissue if samples are available to help validate the genotypes and smad3 expression.

Thank you for this comment. We were not able perform quantitave PCR.

Which part of the plaque are we looking at in Figure 1? It is helpful to infer the function of SMAD3 in certain cell types.

Thank you for this comment. The figure was revised; small magnification (overview) and large magnification (detail). We are looking at the inner side of the sequester, i.e. the part that is looking to the blood (intima + inner part of media). During surgical procedure, the surgeon performs incision in the intima, and that it goes on and removes intima and part of media where atherosclerotic stenotic lesion lies.

Statistical analysis of Figure 1 results should be presented as well. It is important to mention how many patient samples are included for SMAD3 staining and what stages of carotid atherosclerosis they are at.

Thank you for this comment. We revised the paper as suggested, and revised the figure; we added small magnification (overview) and large magnification (detail). Moreover, we added the needed info in the Methods section (2.3. Paragraph):

Chapter 2.3.:

Immunohistochemical analysis was performed on 26 endarterectomy specimens obtained from 26 paticipants with advanced carotid atherosclerosis during carotid surgery

Reviewer 2 Report

Comments and Suggestions for Authors

The study aimed at investigating the potential association between the rs17228212 polymorphism of the SMAD3 gene and the presence of advanced carotid atherosclerosis in a relatively small cohort of Slovenian subjects. The topic is of potential interest, however the manuscript should be carefully revised and amended for English, since several sentences are unclear and need to be reworded. 

More detailed data about the main results of the study might be presented in the proper section of the abstract, which appears quite generic. 

Comments on the Quality of English Language

English is poor and should be improved. 

Author Response

The study aimed at investigating the potential association between the rs17228212 polymorphism of the SMAD3 gene and the presence of advanced carotid atherosclerosis in a relatively small cohort of Slovenian subjects. The topic is of potential interest, however the manuscript should be carefully revised and amended for English, since several sentences are unclear and need to be reworded. 

More detailed data about the main results of the study might be presented in the proper section of the abstract, which appears quite generic. 

Thank you for these comments. We revised the English language. Moreover, we added some details in the Result section as suggested.

Comments on the Quality of English Language:

We revised the English language.

English is poor and should be improved. 

Thank you for this comment. We revised the language as suggested.

Reviewer 3 Report

Comments and Suggestions for Authors

The authors have investigated one genomic polymorphism of SMAD3 in the Slovenian population and its association with SMAD3 protein level in carotid plaques. There are some concerns regarding the results:

1. From Table 1, the control group seems to have more atherosclerotic risk factors(diabetes, more males, higher LDL, etc), I suggest a further discussion on why these subjects don't develop carotid plaques.

2. The histology images are not fully informative, better to provide a zoom-out whole plaque section staining to show total SMAD3 levels  in different SNP genotype subjects with quantification figure, as well as SMC and macrophage marker IHC,  which is associated with plaque morphology and stability related to SMAD3 and TGFB pathways.

3. The authors' hypothesis is that SMAD3 regulates TGFB expression, then how about the plasma circulating TGFB1 level among these 3 genotype populations? Or to check TGFB protein level in the plaques with IHC.

minor: 

References 30,31,32 need editing.

Author Response

The authors have investigated one genomic polymorphism of SMAD3 in the Slovenian population and its association with SMAD3 protein level in carotid plaques. There are some concerns regarding the results:

  1. From Table 1, the control group seems to have more atherosclerotic risk factors(diabetes, more males, higher LDL, etc), I suggest a further discussion on why these subjects don't develop carotid plaques.

Thank you for this comment. We agree that the control group has a lot of risk factors, and that it is not optimal in this regard.

The histology images are not fully informative, better to provide a zoom-out whole plaque section staining to show total SMAD3 levels  in different SNP genotype subjects with quantification figure, as well as SMC and macrophage marker IHC,  which is associated with plaque morphology and stability related to SMAD3 and TGFB pathways.

Thank you for this comment. We provided the whole plaque section (small magnification) as well as detail of the plaques (high magnification). We have available only SMAD3 set for immunostaining.

The authors' hypothesis is that SMAD3 regulates TGFB expression, then how about the plasma circulating TGFB1 level among these 3 genotype populations? Or to check TGFB protein level in the plaques with IHC.

Thank you for this comment. We do not have immunohistochemical set for the TGFB expression at the moment. We ordered it, but it takes time before we get it and perform staining. Due to that fact we were not able to add  TGFB expression in our revised version

minor: 

References 30,31,32 need editing.

Thank you for this comment. We revised the references as suggested.

Reviewer 4 Report

Comments and Suggestions for Authors

lines 58-62, overcrowded sentence, hard to perceive

lines 62-63, few specialists would agree with that statement. More accepted is the view of citations 18-20

line 81 In the Introduction some explanation should be given why this particular genetic polymorphism has been chosen for investigation.

lines 144-145 A more detailed description is needed on the endarterectomies performed. What specimens have been removed?

lines 154-156  It is not clear whether positive cell numbers or pooled  positive cell areas have been  taken into consideration. Also, cell-free areas in endarterectomy specimens can affect the parameters which depend on the applied surgery technique and the selected location in the specimen.

line 231 dedifferentiated

line 244 Our results are in accordance with theirs….

line 255       cell growth inhibition?

 There were  - as it could be expected – significant health problems in the so called “control” group (Diabetes, hypertension, metabolic syndrome etc.) Their evaluation in the Discussion can be hardly avoided.

line 266 The role of TGFbeta in inflammatory phase transition of endothelial and vascular smooth muscle cells, thus in initiation and development of atherosclerotic plaques has been thoroughly proven and is widely accepted. The study performed here is just not providing any opposite proof. However, taking into consideration TGFbeta’s role in formation of collagenous fibers it can play some positive role in stabilizing an existing plaque (See e. g. Ohyama Y et al . J Atheroscler Thromb 2012;19:23.)  Discussion should be worded  not to confront with these statements.

Introduction and  Discussion should concentrate on SMAD3 ‘s role in atherosclerosis, TGFbeta function was not studied, it has to be only mentioned. Views on TGFbeta’s role in atherosclerosis, how it is  influenced by SMAD3 should be given in a more balanced manner, one sided description without proofs in own investigative work should be avoided.

Author Response

lines 58-62, overcrowded sentence, hard to perceive

Thank you for this comment. We revised the text as suggested

lines 62-63, few specialists would agree with that statement. More accepted is the view of citations 18-20

Thank you for this comment. We revised the text in the Introduction section.

line 81 In the Introduction some explanation should be given why this particular genetic polymorphism has been chosen for investigation.

Thank you for this comment. The SMAD3 gene was chosen due to its potential pathogenetic role in the atherosclerotic process.

With regard to the chosen rs, the rs17228212 polymorphism was selected because the minor allele frequency was >10% in the European population.

lines 144-145 A more detailed description is needed on the endarterectomies performed. What specimens have been removed?

In this procedure (endarterectomy) vascular surgeon removes intima and inner part of media of carotid artery. The location of stenotic lesion may be in internal carotid artery or in common carotid artery. In advanced carotid atherosclerosis with significant stenosis we can perform either carotid stenting or surgery.

lines 154-156  It is not clear whether positive cell numbers or pooled  positive cell areas have been  taken into consideration. Also, cell-free areas in endarterectomy specimens can affect the parameters which depend on the applied surgery technique and the selected location in the specimen.

line 231 dedifferentiated ?

line 244 Our results are in accordance with theirs….

Thank you for this comment. We revised the text as suggested

line 255       cell growth inhibition?

Thank you for this comment. We revised the text as suggested

 There were  - as it could be expected – significant health problems in the so called “control” group (Diabetes, hypertension, metabolic syndrome etc.) Their evaluation in the Discussion can be hardly avoided.

Thank you for this comment. We revised the text as suggested

line 266 The role of TGFbeta in inflammatory phase transition of endothelial and vascular smooth muscle cells, thus in initiation and development of atherosclerotic plaques has been thoroughly proven and is widely accepted. The study performed here is just not providing any opposite proof. However, taking into consideration TGFbeta’s role in formation of collagenous fibers it can play some positive role in stabilizing an existing plaque (See e. g. Ohyama Y et al . J Atheroscler Thromb 2012;19:23.)  Discussion should be worded  not to confront with these statements.

Thank you for this comment. We revised the text as suggested.

 Introduction and  Discussion should concentrate on SMAD3 ‘s role in atherosclerosis, TGFbeta function was not studied, it has to be only mentioned. Views on TGFbeta’s role in atherosclerosis, how it is  influenced by SMAD3 should be given in a more balanced manner, one sided description without proofs in own investigative work should be avoided.

Thank you for this comment. We revised the text as suggested, and hopefully it is now more balanced

Round 2

Reviewer 2 Report

Comments and Suggestions for Authors

No further comments

Comments on the Quality of English Language

No further comments

Author Response

The language was reviewed by the expert and revised when needed.

Reviewer 3 Report

Comments and Suggestions for Authors

1. The authors have not answered some of my major concerns. If the authors can't do more IHC staining, then suggest providing 26 samples' SMAD3 staining positive cell quantification figures based on the SNPs.

2. The explanation for the baseline parameter imbalance between the control and case group is not convincing enough, the author assumed it due to "A possible explanation is that the participants in the case group were 235 more compliant with regards to their therapy."  Studies have reported that Statin treatment can affect TGFb-SMAD3 signaling, suggest adding Statin treatment information and regression analysis. 

Author Response

  1. The authors have not answered some of my major concerns. If the authors can't do more IHC staining, then suggest providing 26 samples' SMAD3 staining positive cell quantification figures based on the SNPs.

We have provided photos of all the 26 samples that were immunohistochemically stained for SMAD3.

  1. The explanation for the baseline parameter imbalance between the control and case group is not convincing enough, the author assumed it due to "A possible explanation is that the participants in the case group were 235 more compliant with regards to their therapy."  Studies have reported that Statin treatment can affect TGFb-SMAD3 signaling, suggest adding Statin treatment information and regression analysis. 

We have additionally obtained data regarding the statin treatment from the medical records of the participants and included them in our analysis.

Reviewer 4 Report

Comments and Suggestions for Authors

Questions raised by this reviewer have been answered,  corresponding alterations in the text have been performed.

Author Response

Thank you for your comments.

Additionally, some additional info was added as suggested by the reviewer 3

Round 3

Reviewer 3 Report

Comments and Suggestions for Authors

The authors have revised and improved the data presentation, with just one minor suggestion add one quantification figure bar in Figures 1 and 2.

Author Response

Paper was revised as suggested - we added details to the figures 1 and 2